# To ArXiv or not to ArXiv: A Study Quantifying Pros and Cons of Posting Preprints Online

## Abstract

Double-blind conferences have engaged in debates over whether to allow authors to post their papers online on arXiv or elsewhere during the review process. Independently, some authors of research papers face the dilemma of whether to put their papers on arXiv due to its pros and cons. We conduct a study to substantiate this debate and dilemma via quantitative measurements. Specifically, we conducted surveys of reviewers in two top-tier double-blind computer science conferences—ICML 2021 (5361 submissions and 4699 reviewers) and EC 2021 (498 submissions and 190 reviewers). Our two main findings are as follows. First, more than a third of the reviewers self-report searching online for a paper they are assigned to review. Second, outside the review process, we find that preprints from better-ranked affiliations see a weakly higher visibility, with a correlation coefficient of 0.06 in ICML and 0.05 in EC. In particular, papers associated with the top-10-ranked affiliations had a visibility of approximately 11% in ICML and 22% in EC, whereas the remaining papers had a visibility of 7% and 18% respectively.

## 1 Introduction

Across academic disciplines, peer review is used to decide on the outcome of manuscripts submitted for publication. Single-blind reviewing used to be the predominant method in peer review, where the authors' identities are revealed to the reviewer in the submitted paper. However, several studies have found various biases in single-blind reviewing. These include bias pertaining to affiliations or fame of authors (Blank, 1991; Sun et al., 2021; Manzoor & Shah, 2021), bias pertaining to gender of authors (Rossiter, 1993; Budden et al., 2008; Knobloch-Westerwick et al., 2013; Roberts & Verhoef, 2016), and others (Link, 1998; Snodgrass, 2006; Tomkins et al., 2017). These works span several fields of science and study the effect of revealing the authors' identities to the reviewer through both observational studies and randomized control trials. These biases are further exacerbated due to the widespread prevalence of the Matthew effect—rich get richer and poor get poorer—in academia (Merton, 1968; Squazzoni & Claudio, 2012; Thorngate & Chowdhury, 2013). Biases based on authors' identities in peer review, coupled with the Matthew effect, can have far-reaching consequences on researchers' career trajectories.

As a result, many peer-review processes have moved to *double-blind* reviewing, where authors' names and other identifiers are removed from the submitted papers. Ideally, in a double-blind review process, neither the authors nor the reviewers of any papers are aware of each others' identity. However, a challenge for ensuring that reviews are truly double-blind is the exponential growth in the trend of posting papers online before review (Xie et al., 2021). Increasingly, authors post their preprints on online publishing websites such as arXiv and SSRN and publicize their work on social media platforms such as Twitter. The conventional publication route via peer review is infamously long and time-consuming. On the other hand, online preprint-publishing venues provide a platform for sharing research with the community usually without delays. Not only does this help science move ahead faster, but it also helps researchers avoid being "scooped". However, the increase in popularity of making papers publicly available—with author identities—before or during the

review process, has led to the dilution of double-blinding in peer review. For instance, the American Economic Association, the flagship journal in economics, dropped double-blinding in their reviewing process citing its limited effectiveness in maintaining anonymity. The availability of preprints online presents a challenge in double-blind reviewing, which could lead to biased evaluations for papers based on their authors' identities, similar to single-blind reviewing.

This dilution has led several double-blind peer-review venues to debate whether authors should be allowed to post their submissions on the Internet, before or during the review process. For instance, top-tier machine learning conferences such as NeurIPS and ICML do not prohibit posting online. On the other hand, the Association of Computational Linguistics (ACL) recently introduced a policy for its conferences in which authors are prohibited from posting their papers on the Internet starting a month before the paper submission deadline till the end of the review process. The Conference on Computer Vision and Pattern Recognition (CVPR) has banned the advertisement of submissions on social media platforms for such a time period. Some venues are stricter, for example, the IEEE Communication Letters and IEEE International Conference on Computer Communications (INFOCOMM) disallows posting preprints to online publishing venues before acceptance.

Independently, authors who perceive they may be at a disadvantage in the review process if their identity is revealed face a dilemma regarding posting their work online. On one hand, if they post preprints online, they they are at risk of de-anonymization in the review process, if their reviewer searches for their paper online. Past research suggests that if such de-anonymization happens, reviewers may get biased by the author identity, with the bias being especially harmful for authors from less prestigious organizations. On the other hand, if they choose to not post their papers online before the review process, they stand to lose out on viewership and publicity for their papers.

It is thus important to quantify the consequences of posting preprints online to (i) enable an evidence-based debate over conference policies, and (ii) help authors make informed decisions about posting preprints online. In our work, we conduct a large-scale survey-based study in conjunction with the review process of two top-tier publication venues in computer science that have double-blind reviewing: the 2021 International Conference on Machine Learning (ICML 2021) and the 2021 ACM Conference on Economics and Computation (EC 2021).[1] Specifically, we design and conduct experiments aimed at answering the following research questions:

(Q1) What fraction of reviewers, who had not seen the paper they were reviewing before the review process, deliberately search for the paper on the Internet during the review process?

(Q2) Given a preprint is posted online, what is the causal effect of the rank of the authors' affiliations on the visibility of a preprint to its target audience?

Our work can help inform authors' choices of posting preprints as well as enable evidence-based debates and decisions on conference policies. By addressing these research questions, we aim to measure some of the effects of posting preprints online, and help quantify their associated risks and benefits for authors with different affiliations. Combined, the two research questions tell authors from different institutions how much reach and visibility their preprint gets outside the review process, and at the same time, how likely is their paper to be searched online by reviewers during the review process. These takeaways will help authors trade off the two outcomes of posting preprints online, according to the amount of emphasis they want to place on the pro and the con. Our results also inform conference policies. Specifically, our results for Q1 suggest explicitly instructing reviewers to refrain from searching for their assigned papers online. Our results for Q2 help supplement debates in conferences about allowing preprints, that were previously primarily driven by opinions, with actual data. The data collected in Q2 shows authors preprint posting habits stratified by time, and also shows the amount of visibility the preprints get from their target audience.

---

[1]In Computer Science, conferences are typically the terminal publication venue and are typically ranked at par or higher than journals. Full papers are reviewed in CS conferences, and their publication has archival value.

Further, through our investigation of preprint posting behavior and the viewership received by these preprints, we provide data-driven evidence of trends therein. Specifically, our analysis informs on the fraction of papers made available online before review, how preprint-posting behaviors vary across authors from different affiliations, and the average rate of views obtained by a preprint from researchers in its target audience.

Finally, we list the main takeaways from our research study and analysis:

- In double-blind review processes, reviewers should be explicitly instructed to not search for their assigned papers on the Internet.

- For posted preprints online, authors from lower-ranked institutions enjoy only marginally lower visibility for their papers than authors from top-ranked institutions, implying that authors from lower-ranked institutions may expect almost similar benefits of posting preprints online.

- On average, authors posting preprints online receive viewership from 8% of relevant researchers in ICML and 20% in EC before the conclusion of the review process.

- Certain conferences ban posting preprints a month before the submission deadline. In ICML and EC (which did not have such a ban), more than 50% of preprints posted online where posted before the 1 month period, and these enjoyed a visibility of 8.6% and 18% respectively.

- Conference policies designed towards banning authors from publicising their work on social media or posting preprints before the review process may have only limited effectiveness in maintaining anonymity.

## 2  Related work

*Surveys of reviewers.* Several studies survey reviewers to obtain insights into reviewer perceptions and practices. Nobarany et al. (2016) surveyed reviewers in the field of human-computer interaction to gain a better understanding of their motivations for reviewing. They found that encouraging high-quality research, giving back to the research community, and finding out about new research were the top general motivations for reviewing. Along similar lines, Tite & Schroter (2007) surveyed reviewers in biomedical journals to understand why peer reviewers decline to review. Among the respondents, they found the most important factor to be conflict with other workload.

Resnik et al. (2008) conducted an anonymous survey of researchers at a government research institution concerning their perceptions about ethical problems with journal peer review. They found that the most common ethical problem experienced by the respondents was incompetent review. Additionally, 6.8% respondents mentioned that a reviewer breached the confidentiality of their article without permission. This survey focused on the respondents' perception, and not on the actual frequency of breach of confidentiality. In another survey, by Martinson et al. (2005), 4.7% authors self-reported publishing the same data or results in more than one publication. Fanelli (2009) provides a systematic review and meta analysis of surveys on scientific misconduct including falsification and fabrication of data and other questionable research practices.

Goues et al. (2018) surveyed reviewers in three double-blind conferences to investigate the effectiveness of anonymization of submitted papers. In their experiment, reviewers were asked to guess the authors of the papers assigned to them. Out of all reviews, 70%-86% of the reviews did not have any author guess. Here, absence of a guess could imply that the reviewer did not have a guess or they did not wish to answer the question. Among the reviews containing guesses, 72%-85% guessed at least one author correctly.

*Analyzing papers posted versus not posted on arXiv.* Bharadhwaj et al. (2020) aim to analyse the risk of selective de-anonymization through an observational study based on open review data from the International Conference on Learning Representations (ICLR). The analysis quantifies the risk of de-anonymization by computing the correlation between papers' acceptance rates and their authors' reputations separately for

papers posted and not posted online during the review process. This approach however is hindered by the confounder that the outcomes of the analysis may not necessarily be due to de-anonymization of papers posted on arXiv, but could be a result of higher quality papers being selectively posted on arXiv by famous authors. Moreover, it is not clear how the paper draws conclusions based on the analysis presented therein. Our supporting analysis overlaps with the investigation of Bharadhwaj et al. (2020): we also investigate the correlation between papers' acceptance rates and their authors' associated ranking in order to support our main analysis and to account for confounding by selective posting by higher-ranked authors.

Aman (2014) investigate possible benefits of publishing preprints on arXiv in *Quantitative Biology*, wherein they measure and compare the citations received by papers posted on arXiv and those received by papers not posted on arXiv. A similar confounder arises here that a positive result could be a false alarm due to higher quality papers being selectively posted on arXiv by authors. Along similar lines, Feldman et al. (2018) investigate the benefits of publishing preprints on arXiv selectively for papers that were accepted for publication in top-tier CS conferences. They find that one year after acceptance, papers that were published on arXiv before the review process have 65% more citations than papers posted on arXiv after acceptance. The true paper quality is a confounder in this analysis as well.

In our work, we quantify the risk of de-anonymization by directly studying reviewer behaviour regarding searching online for their assigned papers. We quantify the effects of publishing preprints online by measuring their visibility using a survey-based experiment querying reviewers whether they had seen a paper before.

*Studies on peer review in computer science.* Our study is conducted in two top-tier computer science conferences and contributes to a growing list of studies on peer review in computer science. Lawrence & Cortes (2014); Beygelzimer et al. (2021) quantify the (in)consistencies of acceptance decisions on papers. Several papers (Madden & DeWitt, 2006; Tung, 2006; Tomkins et al., 2017; Manzoor & Shah, 2021) study biases due to single-blind reviewing. Shah et al. (2018) study several aspects of the NeurIPS 2016 peer-review process. Stelmakh et al. (2021c) study biases arising if reviewers know that a paper was previously rejected. Stelmakh et al. (2021b) study a pipeline for getting new reviewers into the review pool. Stelmakh et al. (2020) study herding in discussions. Stelmakh et al. (2022) study citation bias in peer review. A number of recent works (Charlin & Zemel, 2013; Stelmakh et al., 2021a; Kobren et al., 2019; Jecmen et al., 2020; Noothigattu et al., 2021) have designed algorithms that are used in the peer-review process of various computer science conferences. See Shah (2021) for an overview of such studies and computational tools to improve peer review.

## 3 Methods

We now outline the design of the experiment that we conducted to investigate the research questions in this work. First, in Section 3.1 we introduce the two computer science conferences ICML 2021 and EC 2021 that formed the venues for our investigation, and describe research questions Q1 and Q2 in the context of these two conferences. Second, in Section 3.2 we describe the experimental procedure. Finally, in Section 3.3 we provide the details of our analysis methods.

### 3.1 Preliminaries

**Experiment setting.** The study was conducted in the peer-review process of two conferences:

- **ICML 2021** International Conference on Machine Learning is a flagship machine learning conference. ICML is a large conference with 5361 submissions and 4699 reviewers in its 2021 edition.

- **EC 2021** ACM Conference on Economics and Computation is the top conference at the intersection of Computer Science and Economics. EC is a relatively smaller conference with 498 submissions and 190 reviewers in its 2021 edition.

Importantly, the peer-review process in both conferences, ICML and EC, is organized in a double-blind manner, defined as follows. In a **double-blind peer-review process**, the identity of all the authors is removed from the submitted papers. No part of the authors' identity, including their names, affiliations, and seniority, is available to the reviewers through the review process. At the same time, no part of the reviewers' identity is made available to the authors through the review process.

We now formally define some terminology used in the research questions Q1 and Q2. The first research question, Q1, focuses on the fraction of reviewers who deliberately search for their assigned paper on the Internet. The second research question, Q2, focuses on the correlation between the visibility to a target audience of papers available on the Internet before the review process, and the rank of the authors' affiliations. In what follows, we explicitly define the terms used in Q2 in the context of our experiments—target audience, visibility, preprint, and rank associated with a paper.

**Paper's target audience.** For any paper, we define its target audience as members of the research community that share similar research interests as that of the paper. In each conference, a 'similarity score' is computed between each paper-reviewer pair, which is then used to assign papers to reviewers. We used the same similarity score to determine the target audience of a paper (among the set of reviewers in the conference). We provide more details in Appendix A.

**Paper's visibility.** We define the visibility of a paper to a member of its target audience as a binary variable which is 1 if that person has seen this paper outside of reviewing contexts, and 0 otherwise. Visibility, as defined here, includes reviewers becoming aware of a paper through preprint servers or other platforms such as social media, research seminars and workshops. On the other hand, visibility does *not* include reviewers finding a paper during the review process (e.g., visibility does not include a reviewer discovering an assigned paper by deliberate search or accidentally while searching for references).

**Preprint.** To study the visibility of papers released on the Internet before publication, we checked whether each of the papers submitted to the conference was available online. Specifically, for EC, we manually searched for all submitted papers to establish their presence online. On the other hand, for ICML, owing to its large size, we checked whether a submitted paper was available on arXiv (`arxiv.org`). ArXiv is the predominant platform for pre-prints in machine learning; hence we used availability on arXiv as a proxy indicator of a paper's availability on the Internet.

**Rank associated with a paper.** In this paper, the rank of an author's affiliation is a measure of author's prestige that, in turn, is transferred to the author's paper. We determine the rank of affiliations in ICML and EC based on widely available rankings of institutions in the respective research communities. Specifically, in ICML, we rank (with ties) each institution based on the number of papers published in the ICML conference in the preceding year (2020) with at least one author from that institution (Ivanov, 2020). On the other hand, since EC is at the intersection of two fields, economics and computation, we merge three rankings—the QS ranking for computer science (QS, 2021a), the QS ranking for economics and econometrics (QS, 2021b), and the CS ranking for economics and computation (CSRankings, 2021)—by taking the best available rank for each institution to get our ranking of institutions submitting to EC. By convention, better ranks, representing more renowned institutions, are represented by lower numbers; the top-ranked institution for each conference has rank 1. Finally, we define the rank of a paper as the rank of the best-ranked affiliation among the authors of that paper. Due to ties in rankings, we have 37 unique rank values across all the papers in ICML 2021, and 66 unique rank values across all the papers in EC 2021.

## 3.2 Experiment design

To address Q1 and Q2, we designed survey-based experiments for EC 2021 and ICML 2021, described next.

**Design for Q1.** To find the fraction of reviewers that deliberately search for their assigned paper on the Internet, we surveyed the reviewers. Importantly, as reviewers may not be comfortable answering questions about deliberately breaking the double-blindness of the review process, we designed the survey to be anonymous. We used the Condorcet Internet Voting Service (CIVS) (Myers, 2003), a widely used service to conduct secure and anonymous surveys. Further, we took some steps to prevent our survey from spurious responses (e.g., multiple responses from the same reviewer). For this, in EC, we generated a unique link for each reviewer that accepted only one response. In ICML we generated a link that allowed only one response per IP address and shared it with reviewers asking them to avoid sharing this link with anyone.[2] The survey form was sent out to the reviewers via CIVS after the initial reviews were submitted. In the e-mail, the reviewers were invited to participate in a one-question survey on the consequences of publishing preprints online. The survey form contained the following question:

> "During the review process, did you search for any of your assigned papers on the Internet?"

with two possible options: *Yes* and *No*. The respondents had to choose exactly one of the two options. To ensure that the survey focused on reviewers deliberately searching for their assigned papers, right after the question text, we provided additional text: "Accidental discovery of a paper on the Internet (e.g., through searching for related works) does not count as a positive case for this question. Answer *Yes* only if you tried to find an assigned paper itself on the Internet."

Following the conclusion of the survey, CIVS combined the individual responses, while maintaining anonymity, and provided the total number of *Yes* and *No* responses received.

**Design for Q2.** Recall that for Q2 we want to find the effect of a preprint's associated rank on its visibility to a target audience. Following the definitions provided in Section 3.1, we designed a survey-based experiment as follows. We conducted a survey to query reviewers about some papers for which they are considered a target audience. Specifically, we asked reviewers if they had seen these papers before outside of reviewing contexts. We provide more details about the survey, including the phrasing of the survey question, in Appendix A. We queried multiple reviewers about each paper, and depending on their response, we considered the corresponding visibility to be 1 if the reviewer said they had seen the paper before outside of reviewing contexts and 0 otherwise. We note that in ICML reviewers were queried about the papers they were assigned to review using the reviewer response form, in which a response to the question of visibility was required. Meanwhile, in EC, reviewers were queried about a set of papers that they were not assigned to review, using a separate optional survey form that was emailed to them by the program chairs after the rebuttal phase and before the announcement of paper decisions. The survey designed for Q2 had a response rate of 100% in ICML, while EC had a response rate of 55.78%.

### 3.3 Analysis

We now describe the analysis for the data collected to address Q1 and Q2. Importantly, our analysis is the same for the data collected from ICML 2021 and EC 2021. For Q1, we directly report the numbers obtained from CIVS regarding the fraction of reviewers who searched for their assigned papers online in the respective conference. In the rest of this section, we describe our analysis for Q2, where we want to identify and analyse the effect of papers' ranking on their visibility. Recall that for Q2, we collected survey responses and data about papers submitted to ICML or EC that were posted online before the corresponding review process. Since the data is observational, and Q2 aims to identify the causal effect, we describe the causal model assumed in our setting and the interactions therein in Section 3.3.1 followed by the corresponding analysis procedure in Section 3.3.2 and additional supporting analysis in Section 3.3.3.

#### 3.3.1 Graphical causal model for Q2

---

[2]The difference in procedures between EC and ICML is due to a change in the CIVS policy that was implemented between the two surveys.

In Figure 1, we provide the graphical causal model assumed in our setting. To analyse the direct causal effect of a paper's associated ranking (denoted by $\mathbf{R}$) on the visibility (denoted by $\mathbf{V}$) enjoyed by the paper online from its target audience, we consider the interactions in the graphical causal model in Figure 1, which captures three intermediate factors: (1) whether the preprint was posted online, denoted by $\mathbf{P}$, (2) the amount of time for which the preprint has been available online, denoted by $\mathbf{T}$, and (3) the objective quality of the paper, denoted by $\mathbf{Q}$. We now provide an explanation for this causal model.

First, the model captures mediation of effect of $\mathbf{R}$ on $\mathbf{V}$ by the amount of time for which the preprint has been available online, denoted by $\mathbf{T}$. For a paper posted online, the amount of time for which it has been available on the Internet can affect the visibility of the paper. For instance, papers posted online well before the deadline may have higher visibility as compared to papers posted near the deadline. Moreover, the time of posting a paper online could vary across institutions ranked differently. Thus, amount of time elapsed since posting can be a mediating factor causing indirect effect from $\mathbf{R}$ to $\mathbf{V}$. This is represented in Figure 1 by the causal pathway between $\mathbf{R}$ and $\mathbf{V}$ via $\mathbf{T}$.

Second, in the causal model, we consider the papers not posted online before the review process. For a preprint not posted online, we have $\mathbf{T} = 0$, and we do not observe its visibility. However, it is of interest that the choice of posting online before or after the review process could vary depending on both the quality of the paper as well as the rank of the authors' affiliations. For instance, authors from lower-ranked institutions may refrain from posting preprints online due to the risk of de-anonymization in the review process. Or, they may selectively post their high quality papers online. This is captured in our model by introducing the variable $\mathbf{P}$. In Figure 1, this interaction is captured by the direct pathway from $\mathbf{R}$ to $\mathbf{P}$ and the pathway from $\mathbf{R}$ to $\mathbf{P}$ via $\mathbf{Q}$.

Finally, we explain the missing edges in our causal model. In the model, we assume that there is no causal link between $\mathbf{Q}$ and $\mathbf{V}$, this assumes that the initial viewership achieved by a paper does not get effected by its quality. Next, there is no link from $\mathbf{P}$ to $\mathbf{V}$ since the variable $\mathbf{T}$ captures all the information of $\mathbf{P}$. Further, there is no causal link from $\mathbf{Q}$ to $\mathbf{T}$. Here, we assume that $\mathbf{P}$ captures all the information of $\mathbf{Q}$ relevant to $\mathbf{T}$. Lastly, there is no causal link from $\mathbf{Q}$ to $\mathbf{R}$, since the effect of quality of published papers on the rank of the institution would be slow given the infrequent change in ranks.

To address the research question Q2, we want to identify the direct causal effect of a preprint's associated rank $\mathbf{R}$, on its visibility $\mathbf{V}$. Here the term "direct causal effect" is meant to quantify an influence that is not mediated by other variables in the model. Thus, to address Q2 we have to compute the effect of $\mathbf{R}$ on $\mathbf{V}$ through the direct link. In Q2, recall that we consider the papers that were posted online before the review process, which implies $\mathbf{P} = 1$. Since $\mathbf{P}$ is fixed, we do not consider effect from $\mathbf{R}$ to $\mathbf{V}$ via $\mathbf{P}$. Consequently, to compute direct effect of $\mathbf{R}$ on $\mathbf{V}$, we have to account for the indirect causal pathway from $\mathbf{R}$ to $\mathbf{V}$ via $\mathbf{T}$, shown in Figure 1. To control for mediating by $\mathbf{T}$, we provide a detailed description of our estimator in Section 3.3.2.

### 3.3.2  Analysis procedure for Q2

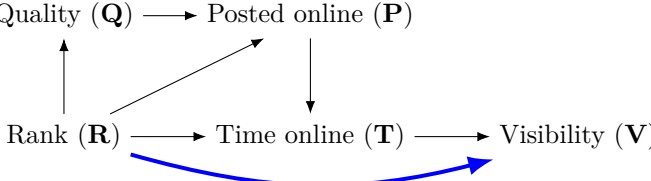

Figure 1: Graphical causal model illustrating the model assumed in our setting, to analyse the direct effect of "Rank" associated with a paper on the "Visibility" enjoyed by the paper as a preprint available on the internet.

We now describe our analysis to identify the direct causal effect of papers' associated ranks on their visibility, in detail. In this analysis, as in the survey for Q2, we consider all the papers submitted to the respective conferences that were posted online before the review process. In other words, the effect identified in this analysis gives the causal effect of papers' rank on visibility under the current preprint-posting habits of authors. In the following analysis procedure, we consider each response obtained in the survey for Q2 as one unit. Each response corresponds to a paper-reviewer pair, wherein the reviewer was queried about seeing the considered paper. In case of no response from reviewer, we do not consider the corresponding paper-reviewer pairs in our data. We thus have two variables associated to each response: the visibility of the paper to the reviewer (in $\{0, 1\}$), and the rank associated with the paper. Recall that we define the rank of a paper as the rank of the best-ranked affiliation associated with that paper.

We first describe the approach to control for mediation by "Time online" in the effect of "Rank" on "Visibility", as shown in Figure 1. There is ample variation in the time of posting papers online within the papers submitted to ICML and EC: some papers were posted right before the review process began while some papers were posted two years prior. To control for the causal effect of time of posting on visibility, we divide the responses into bins based on the number of days between the paper being posted online and the deadline for submitting responses to the Q2 survey. Since similar conference deadlines arrive every three months roughly and the same conference appears every one year, we binned the responses accordingly into three bins. Specifically, if the number of days between the paper being posted online and the survey response is less than 90, it is assigned to the first bin, if the number of days is between 90 and 365, the response is assigned to the second bin, and otherwise, the response is assigned to the third bin. Following this binning, we assume that time of posting does not affect the visibility of papers within the same bin. Consequently, we compute the effect of papers' associated rank on their visibility separately within each bin and then combine them to get the overall effect in two steps:

**Step 1.** We compute the correlation coefficient between papers' visibility and associated rank within each bin. For this we use Kendall's Tau-b statistic, which is closely related to the widely used Kendall's Tau rank correlation coefficient (Kendall, 1938). Kendall's Tau statistic provides a measure of the strength and direction of association between two variables measured on an ordinal scale. It is a non-parametric measure that does not make any assumptions about the data. However, it does not account for ties and our data has a considerable number of ties, since visibility is a binary variable and the rankings used contain ties. Therefore, we use a variant of the statistic, Kendall's Tau-b statistic, that accounts for ties in the data.

Within each bin we consider all the responses obtained and their corresponding visibility and rank value, and compute Kendall's Tau-b correlation coefficient between visibility and rank. The procedure for computing Kendall's Tau-b correlation coefficient between two real-valued vectors (of the same length) is described in Appendix B.1. We now make a brief remark of a notational convention we use in this paper, in order to address ambiguity between the terminology "high-rank institutions" as well as "rank 1, 2,...institutions", both of which colloquially refers to better-rank institutions. It is intuitive to interpret a positive correlation between visibility and rank as the visibility increasing with an *improvement* in the rank. Consequently, we flip the sign of all correlation coefficients computed with respect to the rank variable.

**Step 2.** With the correlation computed within each bin, we compute the overall correlation using a sample-weighted average (Corey et al., 1998). Formally, let $N_1$, $N_2$ and $N_3$ denote the number of responses obtained in the first, second and third bin respectively. Denote Kendall's Tau-b correlation coefficients within the three bins as $\tau_1, \tau_2$ and $\tau_3$. Then the correlation $T$ between papers' visibility and rank over all the time bins is computed as

$$T = \frac{N_1\,\tau_1 + N_2\,\tau_2 + N_3\,\tau_3}{N_1 + N_2 + N_3}. \tag{1}$$

The statistic $T$ gives us the effect size for our research question Q2. Finally, to analyse the statistical significance of the effect, we conduct a permutation test, wherein we permute our data within each bin and

recompute the test statistic $T$ to obtain a $p$-value for our test. We provide the complete algorithm for the permutation test in Appendix B.2.

### 3.3.3 Additional analysis

In this section, we describe our analysis to further understand the relationships between authors' affiliations' ranks and their preprint posting behavior and the quality of the paper. Here we consider all papers submitted to the respective conference.

First we investigate whether the pool of papers posted online before the review process is significantly different, in terms of their rank profile, from the rest of the papers submitted to the conference. Specifically, we analyse the relationship between a binary value indicating whether a submitted paper was posted online before the Q2 survey, and the paper's associated rank. For this, we compute Kendall's Tau-b statistic between the two values for all papers submitted to the conference, and flip the sign of the statistic with respect to the rank variable. This will help us to understand the strength of the causal link from **R** to **P** in Figure 1.

Second, we investigate the causal pathway from **R** to **P** via **Q**. This will help us understand authors' preprint posting habits across different institutions, based on the quality of the preprint. It is of interest to examine whether there is a significant difference between the papers posted and not posted online before the review process, in terms of their quality and rank profile. A key bottleneck in this analysis is that we do not have a handle on the 'quality' of any paper. Thus as a proxy, following past work by Tomkins et al. (2017), we measure the quality of a paper as a binary variable based on its final decision in the conference (`accept` or `reject`). We emphasize this is a significant caveat: the acceptance decisions may not be an objective indicator of the quality of the paper Stelmakh et al. (2019) (for instance, the final decision could be affected by the ranking of the paper in case of de-anonymization of the paper, as discussed in Section 1), and furthermore, identifying an objective quality may not even be feasible (Rastogi et al., 2022).

We conduct this analysis by computing three statistics. First, for all papers posted online before the Q2 survey, we compute Kendall's Tau-b statistic between their rank and their final decision. Second, for all papers *not* posted online, we compute Kendall's Tau-b statistic between their rank and their final decision. Third, for each unique rank value, for the corresponding papers with that rank, we compute the difference between the average acceptance rate for papers posted online and those not posted online. Then, we compute Kendall's Tau-b statistic between the rankings and the difference in acceptance rate. Finally, we flip the sign of all correlation coefficients computed with respect to the rank variable. Hence, a positive correlation would imply that the (difference in) acceptance rate increases as the rank improves.

## 4 Main results

We now discuss the results from the experiments conducted in ICML 2021 and EC 2021.

| | | EC 2021 | ICML 2021 |
|---|---|---|---|
| 1 | # REVIEWERS | 190 | 4699 |
| 2 | # SURVEY RESPONDENTS | 97 | 753 |
| 3 | # SURVEY RESPONDENTS WHO SAID THEY SEARCHED FOR THEIR ASSIGNED PAPER ONLINE | 41 | 269 |
| 4 | % SURVEY RESPONDENTS WHO SAID THEY SEARCHED FOR THEIR ASSIGNED PAPER ONLINE | 42% | 36% |

Table 1: Outcome of survey for research question Q1.

### 4.1 Q1 results

Table 1 provides the results of the survey for research question Q1. The percentage of reviewers that responded to the anonymous survey for Q1 is 16% (753 out of 4699) in ICML and 51% (97 out of 190) in EC. While the coverage of the pool of reviewers is small in ICML (16%), the number of responses obtained is large (753). As shown in Table 1, the main observation is that, in both conferences, at least a third of the Q1 survey respondents self-report deliberately searching for their assigned paper on the Internet. There is substantial difference between ICML and EC in terms of the response rate as well as the fraction of *Yes* responses received, however, the current data cannot provide explanations for these differences.

### 4.2 Q2 results

We discuss the results of the survey conducted for Q2 in ICML 2021 and EC 2021. First we discuss the results of the main analysis described in Section 3.3.2. Then we discuss the results of the additional analysis described in Section 3.3.3. Finally, we discuss other general trends in posting preprints online, and the visibility gained thereof, in ICML 2021 and EC 2021.

**Analysis for Q2.** Table 2 depicts the results of the survey for research question Q2. We received 7594 responses and 449 responses for the survey for Q2 in ICML and EC respectively (Row 1). Recall that in the main analysis, we investigate the effect of papers' associated rank on their visibility, while controlling for mediation by time online, based on the causal model in Figure 1. As shown in Table 2 (Row 4), for papers submitted to the respective conference and posted online before the review process, we find a weak positive effect of the papers' associated rank on their visibility. Here the papers posted online represent the current preprint-posting habits of authors. The weak positive effect implies that the visibility increases slightly as the rank improves.

To provide some interpretation of the correlation coefficient values in Row 4, we compare the mean visibility within and without responses obtained for papers with at least one affiliation ranked 10 or better (Row 8 and 9). There are 10 and 23 institutions among the top-10 ranks in ICML and EC respectively. We see that there is more than 3 percentage points decrease in mean visibility across these two sets of responses in both ICML and EC. Figure 2 displays additional visualization that helps to interpret the strength of the correlation between papers' rank and visibility. The data suggests that top-ranked institutions enjoy higher visibility than lower-ranked institutions in both venues ICML 2021 and EC 2021.

In summary, for papers available online before the review process, in ICML the analysis supports a weak but statistically significant effect of paper ranking on its visibility for preprints available online. In EC the effect size is comparable, but the effect does not reach statistical significance. Without further data, for EC the results are only suggestive.

| | | EC 2021 | ICML 2021 |
|---|---|---|---|
| 1 | # RESPONSES OVERALL | 449 | 7594 |
| 2 | # PAPERS IN BINS 1, 2, 3 | 63, 82, 38 | 968, 820, 146 |
| 3 | # RESPONSES IN BINS 1, 2, 3 | 159, 233, 57 | 3799, 3228, 567 |
| 4 | CORRELATION BETWEEN RANK AND VISIBILITY $[-1, 1]$ | 0.05 ($p = 0.11$) | 0.06 ($p < 10^{-5}$) |
| 5 | CORRELATION BETWEEN RANK AND VISIBILITY IN BINS 1, 2, 3 | 0.06, 0.04, 0.04 | 0.04, 0.10, 0.03 |
| 6 | p-VALUE ASSOCIATED WITH CORRELATIONS IN ROW 5 | 0.36, 0.46, 0.66 | 0.004, $< 10^{-5}$, 0.19 |
| 7 | % VISIBILITY OVERALL $[0 - 100]$ | 20.5% (92 OUT OF 449) | 8.36% (635 OUT OF 7594) |
| 8 | % VISIBILITY FOR PAPERS WITH TOP 10 RANKS $[0 - 100]$ | 21.93% (59 OUT OF 269) | 10.91% (253 OUT OF 2319) |
| 9 | % VISIBILITY FOR PAPERS BELOW TOP 10 RANKS $[0 - 100]$ | 18.33% (33 OUT OF 180) | 7.24% (382 OUT OF 5275) |

Table 2: Outcome of main analysis for research question Q2. A positive correlation in Row 4 and Row 5 implies that the visibility increases as the rank of the paper improves. Recall that for ICML, we consider the set of responses obtained for submissions that were available as preprints on arXiv. There were 1934 such submissions.

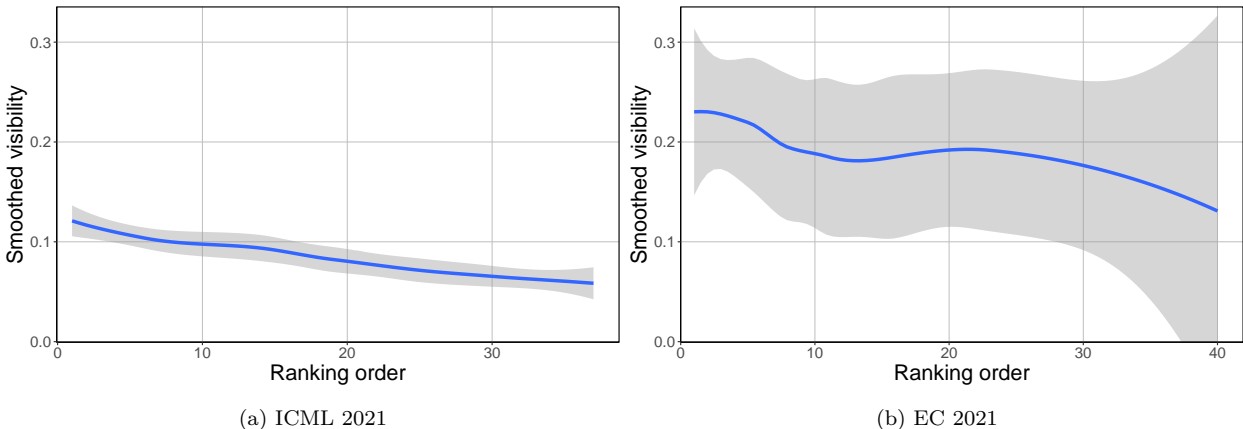

(a) ICML 2021        (b) EC 2021

Figure 2: Using responses obtained in Q2 survey, we plot the papers' visibility against papers' associated rank with smoothing. On the x-axis, we order papers by their ranks (i.e., paper with the best rank gets order 1, paper with the second best rank gets order 2, and so on). The range of x-axis is given by the number of unique ranks in the visibility analysis, which may be smaller than the total number of unique ranks associated with the papers in the respective conferences. The x-axis range is 37 in Figure 2a and 40 in Figure 2b due to ties in rankings used. On the y-axis, smoothed visibility lies in $[0, 1]$. We use local linear regression for smoothing (Cleveland & Loader, 1996). The solid line gives the smoothed visibility, and the grey region around the line gives the 95% confidence interval.

Further, since the survey was optional in EC 2021, we analyse the difference between the responders and non-responders. Specifically, we looked at the distribution of the seniority of the reviewers that did and did not respond to the survey. We measure the seniority of the reviewers based on their reviewing roles, namely, (i) Junior PC member, (ii) Senior PC. member, (iii) Area Chair, valued according to increasing seniority. Based on this measurement system, we investigated the distribution of seniority across the groups of responders and non-responders. We find that there is no significant difference between the two groups, with mean seniority given by 1.64 in the non-responders' group and 1.61 in the responders' group. The difference in the mean (0.03) is much smaller than the standard deviation in each group, which is 0.64 and 0.62 respectively.

**Additional analysis.** We provide the results for the supporting analysis described in Section 3.3.3 in Table 3. Recall that, in this analysis, we consider all papers submitted to the respective conferences. There were a total of 5361 and 498 papers submitted to ICML 2021 and EC 2021 respectively.

Among all the papers submitted, we observe that there is a statistically significant weak positive correlation (Kendall's Tau-b) between paper's rank and whether it was posted online before the review process in both

| | | EC 2021 | ICML 2021 |
|---|---|---|---|
| 1 | # PAPERS | 498 | 5361 |
| 2 | # PAPERS POSTED ONLINE BEFORE THE END OF REVIEW PROCESS | 183 | 1934 |
| 3 | CORRELATION BETWEEN PAPERS' RANK AND WHETHER THEY WERE POSTED ONLINE $_{[-1,1]}$ | 0.09 | 0.12 |
| 4 | CORRELATION FOR PAPERS POSTED ONLINE BETWEEN THEIR RANK AND DECISION $_{[-1,1]}$ | 0.03 | 0.11 |
| 5 | CORRELATION FOR PAPERS NOT POSTED ONLINE BETWEEN THEIR RANK AND DECISION $_{[-1,1]}$ | 0.13 | 0.16 |
| 6 | CORRELATION BETWEEN RANKING AND CORRESPONDING DIFFERENCE, BETWEEN PAPERS POSTED AND NOT POSTED ONLINE, IN MEAN ACCEPTANCE RATE $_{[-1,1]}$ | 0.12 | 0.01 |

Table 3: Outcome of supporting analysis for research question Q2. A positive correlation in rows 3, 4, 5 and 6 implies that the value of the variable considered increases as the rank of the paper improves. For instance, in row 3, the rate of posting online increases as the rank improves.

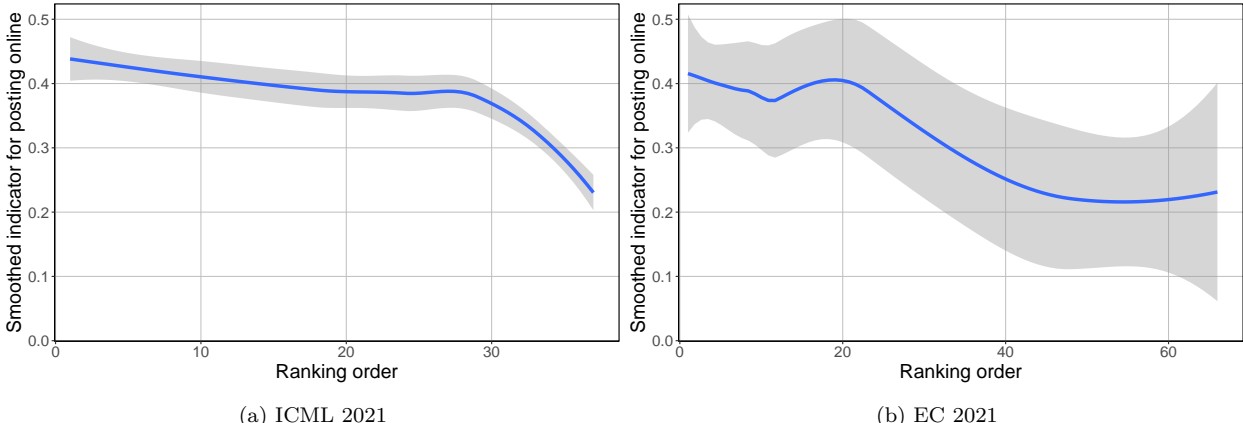

(a) ICML 2021 (b) EC 2021

Figure 3: For papers submitted to the respective conferences, we plot the indicator for paper being posted online before the end of the review process against papers' associated rank, with smoothing. On the x-axis, we have the ranking order as described in Figure 2. On the y-axis, smoothed indicator for posting online lies in $[0, 1]$. We use locally estimated smoothing to get the smoothed indicator for posting online across ranks, shown by the solid line, and a 95% confidence interval, shown by the grey region.

ICML and EC of 0.12 ($p < 10^{-5}$) and 0.09 ($p = 0.01$) respectively (Row 3). This implies that the authors from higher-ranked institutions are more likely to post their papers online before the review process. Further, it provides evidence for a causal link between $\mathbf{R}$ and $\mathbf{P}$ in Figure 1. We provide visualization to interpret the correlation between ranking and posting behaviour in Figure 3.

To understand if there is significant difference in the quality of papers uploaded online by authors from institutions with different ranks, we compare the final decision of the pool of papers posted online before the review process and the pool of papers that was not, across ranks. Now, for the pool of papers posted online, we see that Kendall's Tau-b correlation between papers' rank and final decision is 0.11 ($p < 10^{-5}$) in ICML and 0.03 ($p = 0.58$) in EC (Row 4). Recall that a positive correlation implies that the acceptance rate increases as the rank improves. For the pool of papers *not* posted online, we see that Kendall's Tau-b correlation between papers' rank and final decision, 0.16 ($p < 10^{-5}$) in ICML and 0.13 ($p = 0.006$) in EC (Row 5). Lastly, the correlation between the rank values and the corresponding difference (between papers posted and not posted online) in mean acceptance rates is 0.01 ($p = 0.92$) in ICML and 0.12 ($p = 0.18$) in EC (Row 6).

To interpret these values, we provide visualization of the variation of mean acceptance rate as rank varies for the two pools of papers in Figure 4. Recall that in our assumed causal model in Figure 1, there is a causal link from $\mathbf{R}$ to $\mathbf{P}$ via $\mathbf{Q}$. In ICML (in Figure 4a), we see that there is a clear trend for authors from all institutions posting higher quality papers online as preprints, which implies that quality of the paper mediates the effect of author's rank on their posting decision. Further, we see that the authors from higher-ranked institutes submit papers with a higher acceptance rate, providing evidence for the causal link from $\mathbf{R}$ to $\mathbf{Q}$. Meanwhile, in EC, we see a similar trend for papers not posted online. However, the plot for papers posted online occupies a large region for its 95% confidence intervals, and is thereby difficult to draw insights from.

**Trends in posting preprints online and visibility.** We now discuss the preprint-posting habits of authors, and the viewership received by them from relevant researchers in the community. In Table 3, we see that there were a total of 5361 and 498 papers submitted to ICML 2021 and EC 2021 respectively, out of which 1934 and 183 were posted online before the end of the review process respectively (Row 1 and 2). Thus, we see that more than a third of the papers submitted were available online. Further, based on our binning rule based on time of posting described in Section 3.3.2, we see more papers in bin 1 and bin 2,

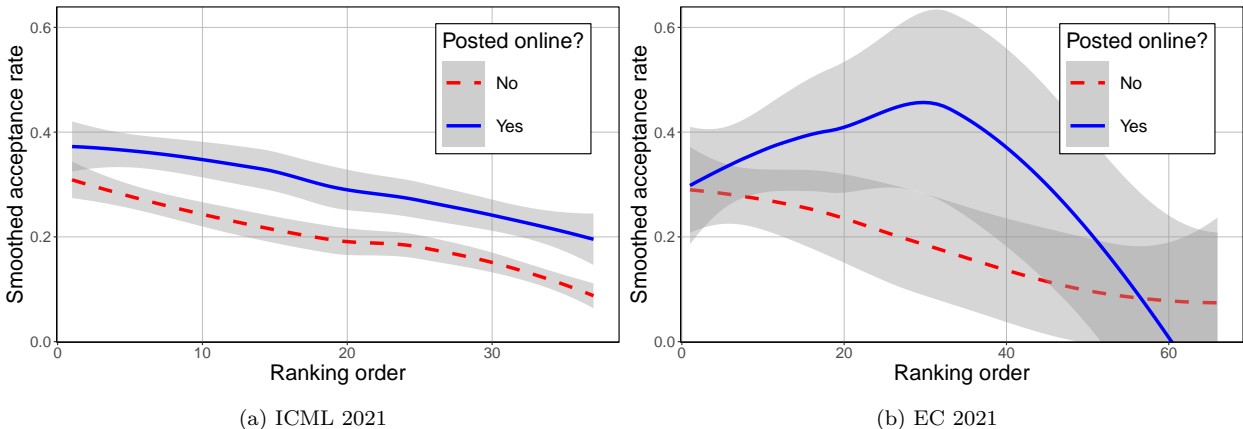

(a) ICML 2021  (b) EC 2021

Figure 4: For papers submitted to the respective conferences and (not) posted online before the review process, we plot the papers' final decision against papers' associated rank, with smoothing. On the x-axis, we have the ranking order as described in Figure 2. On the y-axis, smoothed acceptance rate lies in $[0, 1]$. We use locally estimated smoothing to get the smoothed acceptance rate across ranks, shown by the lines, and a 95% confidence interval, shown by the grey region. Note that in Figure 4b, the number of papers corresponding to the ranks on the right of the plot is very small.

compared to bin 3 (refer Table 2). This suggests that majority of preprints were posted online within one year of the review process (Row 2).

Based on results from Q2 survey, we learn that the mean visibility in ICML 2021 is 8.36% and that in EC 2021 is 20.5% (refer Table 2, Row 7). This provides an estimate of the fraction of relevant researchers in the community viewing preprints available online. Further we note that the mean visibility in ICML is considerably smaller than that in EC. This may be attributed to the following reason: The research community in EC is smaller and more tight-knit, meaning that there is higher overlap in research interests within the members of the community (reviewers). On the other hand, ICML is a large publication venue with a more diverse and spread-out research community.

## 5 Discussion

To improve peer review and scientific publishing in a principled manner, it is important to understand the quantitative effects of the policies in place, and design policies in turn based on these quantitative measurements.

We find that more than a third of survey respondents self-report deliberately searching for their assigned papers online, thereby weakening the effectiveness of author anonymization in double-blind peer review. This finding has important implications for authors who perceive they may be at a disadvantage in the review process if their identity is revealed, in terms of their decision to post preprints online.

Further, the observed value of fraction of reviewers that searched for their assigned paper online in Table 1 might be an underestimate due to two reasons: (i) Reviewers who deliberately broke the double-blindedness of the review process may be more reluctant to respond to our survey for Q1. (ii) As we saw in Section 4.2, roughly 8% of reviewers in ICML 2021 had already seen their assigned paper before the review process began (Table 2 row 5). If these reviewers were not already familiar with their assigned paper, they may have searched for them online during the review process.

Based on the analysis of Q2, we find evidence to support a weak effect of authors' affiliations' ranking on the visibility of their papers posted online before the review process. These papers represent the current preprint-posting habits of authors. Thus, authors from lower-ranked institutions get slightly less viewership

for their preprints posted online compared to their counterparts from top-ranked institutions. For Q2, the effect size is statistically significant in ICML, but not in EC. A possible explanation for the difference is in the method of assigning rankings to institutions, described in Section 3.1. For ICML, the rankings used are directly related to past representation of the institutions at ICML (Ivanov, 2020). In EC, we used popular rankings of institutions such as QS rankings and CS rankings. In this regard, we observe that there is no clear single objective measure for ranking institutions in a research area. This leads to many ranking lists that may not agree with each other. Our analysis also suffers from this limitation. Another possible explanation for the difference is the small sample size in EC.

Next, while we try to carefully account for mediation by time of posting in our analysis for Q2, our study remains dependent on observational data. Thus, the usual caveat of unaccounted for confounding factors applies to our work. For instance, the topic of research may be a confounding factor in the effect of papers' rank on visibility: If authors from better-ranked affiliations work more on cutting-edge topics compared to others, then their papers would be read more widely. This could potentially increase the observed effect.

**Policy implications.** Double-blind venues now adopt various policies for authors regarding posting or advertising their work online before and during the review process. A notable example is a recent policy change by the Association for Computational Linguistics in their conference review process, which includes multiple conferences: ACL, NAACL (North American Chapter of the ACL) and EMNLP (Empirical Methods in Natural Language Processing). ACL introduced an anonymity period for authors, starting a month before the paper submission deadline and extending till the end of the review process. According to their policy, within the anonymity period authors are not allowed to post or discuss their submitted work anywhere on the Internet (or make updates to existing preprints online). In this manner, the conference aims to limit the de-anonymization of papers from posting preprints online. A similar policy change has been instituted by the CVPR computer vision conference. Furthermore, we provide some quantitative insights on this front using the data we collected from the Q2 survey in ICML 2021 and EC 2021. There were 918 (out of 5361 submitted) and 74 (out of 498 submitted) papers posted online *during* the one month period right before the submission deadline in ICML and EC respectively. These papers enjoyed a visibility of 8.11% (292 out of 3600) and 23.81% (45 out of 189) respectively. Meanwhile, there were 1016 (out of 5361) and 109 (out of 498) papers posted online *prior to* the one month period right before the submission deadline in ICML and EC, and these papers enjoyed a visibility of 8.59% (343 out of 3994) and 18.08% (47 out of 260) respectively. Moreover, the combination of the result of the Q1 survey and the finding that a majority of the papers posted online were posted before the anonymity period suggests that conference policies designed towards banning authors from publicising their work on social media or from posting preprints online in a specific period of time before the review process may not be effective in maintaining double anonymity, since reviewers may still find these papers online if available. These measurements may help inform subsequent policy decisions.

While our work finds dilution of anonymization in double-blind reviewing, any prohibition on posting preprints online comes with its own downsides. For instance, consider fields such as Economics where journal publication is the norm, which can often imply several years of lag between paper submission and publication. Double-blind venues must grapple with the associated trade-offs, and we conclude with a couple of suggestions for a better trade-off. First, many conferences, including but not limited to EC 2021 and ICML 2021, do not have clearly stated policies for reviewers regarding searching for papers online, and can clearly state as well as communicate these policies to the reviewers. Second, venues may consider policies requiring authors to use a different title and reword the abstract during the review process as compared to the versions available online, which may reduce the chances of reviewers discovering the paper or at least introduce some ambiguity if a reviewer discovers (a different version of) the paper online.

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

# A  Survey details for Q2.

**Target audience selection**  Recall that our objective in target audience selection is to find reviewers for each paper whose research interests intersect with the paper, so that we can survey these reviewers about having seen the corresponding papers outside of reviewing contexts. We describe the exact process for target audience selection in EC and ICML.

In EC, the number of papers posted online before the end of the review process was small. To increase the total number of paper-reviewer pairs where the paper was posted online and the reviewer shared similar research interests with the paper, we created a new paper-reviewer assignment. For the new paper-reviewer assignment, for each paper we considered at most 8 members of the reviewing committee that satisfied the following constraints as its target audience—(1) they submitted a positive bid for the paper indicating shared interest, (2) they are not reviewing the given paper.

In ICML, a large number of papers were posted online before the end of the review process. So, we did not create a separate paper-reviewer assignment for surveying reviewers. Instead, in ICML, we consider a paper's reviewers as its target audience and queried the reviewers about having seen it, directly through the reviewer response form.

**Survey question.** For research question Q2, we conducted a survey to measure the visibility of papers submitted to the conference and posted online before or during the review process. We describe the details of the survey for EC 2021 and ICML 2021 separately. In EC 2021, we created a specialised reviewer-specific survey form shared with all the reviewers. Each reviewer was shown the title of five papers and asked to answer the following question for each paper:

"Have you come across this paper earlier, outside of reviewing contexts?"

In the survey form, we provided examples of reviewing contexts as "reviewing the paper in any venue, or seeing it in the bidding phase, or finding it during a literature search regarding another paper you were reviewing." The question had multiple choices as enumerated in Table 4, and the reviewer could select more than one choice. If they selected one or more options from (b), (c), (d) and (e), we set the visibility to 1, and if they selected option (a), we set the visibility to 0. We did not use the response in our analysis, if the reviewer did not respond or only chose option (f). In Table 4, we also provide the number of times each choice was selected in the set of responses obtained.

In ICML 2021, we added a two-part question corresponding to the research question Q2 in the reviewer response. Each reviewer was asked the following question for the paper they were reviewing:

"Do you believe you know the identities of the paper authors? If yes, please tell us how."

Each reviewer responded either *Yes* or *No* to the first part of the question. For the second part of the question, table 5 lists the set of choices provided for the question, and a reviewer could select more than one choice. If they responded *Yes* to the first part, and selected one or more options from (a), (d), (e) and (f) for the second part, then we set the visibility to 1, otherwise to 0. In Table 5, we also provide the number of times each choice was selected in the set of responses that indicated a visibility of 1.

## B   Analysis procedure details

In this section we provide some more details of the analysis procedure.

| List of choices for question in Q2 survey | Count |
|---|---|
| (a) I have NOT seen this paper before / I have only seen the paper in reviewing contexts | 359 |
| (b) I saw it on a preprint server like arXiv or SSRN | 51 |
| (c) I saw a talk/poster announcement or attended a talk/poster on it | 22 |
| (d) I saw it on social media (e.g., Twitter) | 4 |
| (e) I have seen it previously outside of reviewing contexts (but somewhere else or don't remember where) | 29 |
| (f) I'm not sure | 24 |

Table 4: Set of choices provided to reviewers in EC in Q2 survey and the number of times each choice was selected in the responses obtained. There were 449 responses in total, out of which 92 responses indicated a visibility of 1.

| List of choices for question in Q2 survey | Count |
|---|---|
| (a) I was aware of this work before I was assigned to review it. | 373 |
| (b) I discovered the authors unintentionally while searching web for related work during reviewing of this paper | 47 |
| (c) I guessed rather than discovered whose submission it is because I am very familiar with ongoing work in this area. | 28 |
| (d) I first became aware of this work from a seminar announcement, Archiv announcement or another institutional source | 259 |
| (e) I first became aware of this work from a social media or press posting by the authors | 61 |
| (f) I first became aware of this work from a social media or press posting by other researchers or groups (e.g. a ML blog or twitter stream) | 52 |

Table 5: Set of choices provided to reviewers in ICML in Q2 survey question and the number of times each choice was selected in the set of responses considered that self-reported knowing the identities of the paper authors outside of reviewing contexts. There were a total of 635 such responses that indicated a visibility of 1. Recall that for ICML, we consider the set of responses obtained for submissions that were available as preprints on arXiv. There were 1934 such submissions.

## B.1 Kendall's Tau-b statistic

We describe the procedure for computing Kendall's Tau-b statistic between two vectors. Let $n$ denote the length of each vector. Let us denote the two vectors as $[x_1, x_2, \ldots, x_n] \in \mathbb{R}^n$ and $[y_1, y_2, \ldots, y_n] \in \mathbb{R}^n$. Let $P$ denote the number of concordant pairs in the two vectors, defined formally as

$$P = \sum_{\substack{(i,k) \in [n]^2 \\ i<k}} \left( \mathbb{I}\left(x_i > x_k\right) \mathbb{I}\left(y_i > y_k\right) + \mathbb{I}\left(x_i < x_k\right) \mathbb{I}\left(y_i < y_k\right) \right).$$

Following this, we let the number of discordant pairs in the two vectors be denoted by $Q$, defined as

$$Q = \sum_{\substack{(i,k) \in [n]^2 \\ i<k}} \left( \mathbb{I}\left(x_i > x_k\right) \mathbb{I}\left(y_i < y_k\right) + \mathbb{I}\left(x_i < x_k\right) \mathbb{I}\left(y_i > y_k\right) \right).$$

Observe that the concordant and discordant pairs do not consider pairs with ties in either of the two vectors. In our data, we have a considerable number of ties. To account for ties, we additionally compute the following statistics. Let $A_x$ and $A_y$ denote the number of pairs in the two vectors tied in exactly one of the two vectors as

$$A_x = \sum_{\substack{(i,k) \in [n]^2 \\ i<k}} \mathbb{I}\left(x_i = x_k\right) \mathbb{I}\left(y_i \neq y_k\right) \qquad \text{and} \qquad A_y = \sum_{\substack{(i,k) \in [n]^2 \\ i<k}} \mathbb{I}\left(x_i \neq x_k\right) \mathbb{I}\left(y_i = y_k\right).$$

Finally, let $A_{xy}$ denote the number of pairs in the two vectors tied in both vectors, as

$$A_{xy} = \sum_{\substack{(i,k) \in [n]^2 \\ i<k}} \mathbb{I}\left(x_i = x_k\right) \mathbb{I}\left(y_i = y_k\right).$$

Observe that the five statistics mentioned above give a mutually exclusive and exhaustive count of pairs of indices, with $P + Q + A_x + A_y + A_{xy} = 0.5n(n-1)$. With this setup in place, we have the Kendall's Tau-b statistic between $[x_1, x_2, \ldots, x_n] \in \mathbb{R}^n$ and $[y_1, y_2, \ldots, y_n] \in \mathbb{R}^n$ denoted by $\tau$ as

$$\tau = \frac{P - Q}{\sqrt{(P + Q + A_x)(P + Q + A_y)}}. \tag{2}$$

This statistic captures the correlation between the two vectors.

> **Input** : Samples $v_i, \alpha_i, \widetilde{t}_i$ for $i \in [N]$, iteration count $\gamma$.
> **(1)** Compute the test statistic $T$ defined in equation 1.
> **(2)** For $z \leftarrow 1$ to $\gamma$:
>    **(i)** For all $b \in \{1, 2, 3\}$: Let $V_b$ denote the number of responses with $v_i = 1$ in bin $b$. Take all the responses in bin $b$ and reassign each response's visibility to 0 or 1 uniformly at random such that the total number of responses with a visibility of 1 remains the same as $V_b$.
>    **(ii)** Using the new values of visibility in all bins, recompute the test statistic in equation 1. Denote the computed test statistic as $T_z$.
> **Output** : $P$ value $= \frac{1}{\gamma} \sum_{z=1}^{\gamma} \mathbb{I}(T_z - T > 0)$.

**Algorithm 1:** Permutation test for correlation between papers' visibility and rank.

## B.2    Permutation test

The test statistic $T$ in equation 1 gives us the effect size for our test. Recall from equation 1 that the test statistic $T$ is defined as:

$$T = \frac{N_1 \, \tau_1 + N_2 \, \tau_2 + N_3 \, \tau_3}{N_1 + N_2 + N_3},$$

where for each bin value $b \in \{1, 2, 3\}$, we have $N_b$ as the number of responses obtained in that bin, and $\tau_b$ represents the Kendall Tau-b correlation between visibility and rank in the responses obtained in that bin. To analyse the statistical significance of the effect, we define some notation for our data. Let $N$ denote the total number of responses. For each response $i \in [N]$ we denote the visibility of the paper to the reviewer as $v_i \in \{0, 1\}$ and the rank associated with response $i$ as $\alpha_i \in \mathbb{N}_{<0}$. Finally, we denote the bin associated with response $i$ as $\widetilde{t}_i \in \{1, 2, 3\}$. With this, we provide the algorithm for permutation testing in Algorithm 1.

