# OpenReview forum: "To ArXiv or not to ArXiv: A Study Quantifying Pros and Cons of Posting Preprints Online"
_TMLR — Rejected by TMLR_

### Review · Reviewer_XT9x · 2022-12-02

**Summary Of Contributions:**

The authors conduct a survey of reviewers for ICML '21 and EC '21 to understand whether reviewers search online for papers they are assigned, and the relationship between affiliation ranking and visibility outside the review process. Statistical analyses are performed on the collected survey data.

**Audience:**

Yes

**Broader Impact Concerns:**

No concerns.

**Claims And Evidence:**

Yes

**Requested Changes:**

- Add some analysis about those reviewers who did not answer the survey at all, comparing their population (somehow) to the population of those that did.
- Consider how a fixed (or random) effects analysis makes sense here. Then either (a) perform a few of these or (b) argue why it's not appropriate or not needed. You can use R or statsmodels.

**Strengths And Weaknesses:**

Thank you to the authors for the hard work on this paper. It's important work and should help policy decisions. Most of my suggestions are about improving Q2 analyses, but the work is already valuable.

Strengths
------------
- Clear writing
- Sensible data collection
- Good motivation
- Extant analyses make sense
- Potential to positively impact policy making for double-blind reviewing and preprints

Weaknesses
----------------
(1) This related paper is perhaps valuable to cite: https://arxiv.org/abs/1805.05238. The authors "observe that papers submitted to arXiv before acceptance have, on average, 65\% more citations in the following year compared to papers submitted after."

(2) You exclude non-answering reviewers from your analysis entirely. Is there anything you can say about those who did not answer the survey? Are there logs of whether they opened the link and abandoned it half-way through? Can you look at their actual reviews and see if they differ systematically in their length, quality, etc? What else do you know about them that can help you understand those reviewers who didn't answer the survey? If they form a very different population, your conclusions become less useful, so it's important to investigate that possibility.

(3) The analysis are piecemeal, but can be made more holistic with a different analytical approach. For this kind of work, I would have expected a mixed-effects (or fixed-effects) multivariate analysis so you can understand multiple factors all at once. For Q2, in section 3.3.2, you note three variables: binary visibility, affiliation-determined rank, and number of days between paper being posted online and survey response. (There's also whether or not the paper got accepted and the review score itself and others you didn't study like the affiliation rank of the reviewer.) Then you bin w.r.t. time of posting and compute the Kendall's Tau-b statistics. A potentially better approach is to perform a fixed effects analysis (something similar is done in https://arxiv.org/abs/1805.05238). For example (expressed in the standard formula language from R):

$\verb|visibility ~ paper_rank + C(time_of_posting)|$,

where $C()$ will bin time_of_posting

With this kind of approach, you can add any number of other variables. For example, whether the paper was accepted could be important and thus:

$\verb|visibility ~ paper_rank + C(time_of_posting) + accepted_or_not|$

Or perhaps it makes more sense to have the acceptance be the output variable here, and assess how much of an effect visibility has on acceptance when accounting for rank and time of posting. And you can throw in reviewer affiliation rank too as it may have interesting relationship with the outcome.

$\verb|accepted_or_not ~ paper_rank + reviewer_rank + C(time_of_posting) + visibility|$

After fitting these models, you would have coefficients with associated p-values as to whether they are statistically significantly different from 0, and these can be reported instead of the Tau-b statistic.

(4) Page 11: "A possible explanation for the difference is in the method of assigning rankings to institutions". I think the more likely and simple explanation is that EC has less data, and that's why the p-values are large. With large-enough sample size, all differences eventually look "significant" in the frequentist framework.

---

### Review · Reviewer_xin1 · 2022-12-24

**Summary Of Contributions:**

This paper reports on a survey of ML conference reviewers and discusses two descriptive findings that bear on the broad question of how arXiv affects the double-blind review process in machine learning conferences. The findings are: 1) around 1/3 of surveyed reviewers at ICML 2021 and EC 2021 self-report searching for their assigned papers online, and 2) preprints by authors from higher-ranked institutions have higher visibility—i.e., were more often already known to reviewers. Much of the paper is dedicated to developing methodology for obtaining the second finding. The proposed methodology seeks to assess a statistical relationship between the preprint’s ranking (of its authors’ institutions) and its visibility among reviewers, which controls for various “confounding” factors like the preprint’s quality or the time the preprint spent online.

**Audience:**

Yes

**Broader Impact Concerns:**

I did not have any broader impact concerns.

**Claims And Evidence:**

No

**Requested Changes:**

If the paper wants to be true to its title, and to ask a causal question, then it must clearly articulate its causal model, defining clearly what the treatment, outcome, and confounding variables and how they relate. It must state exactly what the causal question and estimand are for Q2, and show how the estimator it proposes is a valid/correct estimator for that estimand.

On the other hand, if the paper does not want to commit to a causal question, and wants to simply report descriptive/correlational findings, then it must explain how the bulk of its content on adjusting for “confounders” is relevant and what its estimator that controls for them is actually estimating. I am honestly pessimistic that there would be enough interesting content in this version of the paper for it to merit publication at TMLR, but I am ready to be convinced otherwise.

**Strengths And Weaknesses:**

**Strong motivation & valuable raw data**

This paper studies an important and timely topic in machine learning, namely the state of peer review and the tension between the parallel cultures of preprints and double-blind conference reviewing. Anecdotally, this topic seems to me to be top-of-mind for many in the field, and there seems to be increasing frustration with a lack of coordination and effective policy on the matter. I commend the authors for their initiative to inject some empiricism into the debate. Designing and conducting large-scale surveys is tricky and time-consuming, particularly when there are limited windows of opportunity to deploy them, as with ML conferences that only occur a few times per year. That authors took care to design the survey methodology well and to document it clearly. The community will benefit from the valuable descriptive results afforded by the studies this paper reports as well as from future studies that build on this paper’s methodology.

**To cause or not to cause?**

The paper sets out with two research questions, Q1 and Q2 on page 2, that are both initially phrased to ask about (non-causal) statistical relationships. Q1 simply asks what fraction of reviewers deliberately search for preprints of their assigned papers online. The answer to Q1 is straightforwardly obtained from the survey, which asks this of reviewers directly. The paper asks and answers Q1 clearly and convincingly.

The paper is less clear and convincing in how it asks and answers Q2. The main culprit here the paper’s unwillingness to commit to a causal or non-causal version of Q2. As first formulated on page 2, Q2 meekly asks “What is the **relation** between [rank and visibility]?”. How to answer this relies on what is meant by “relation”, “rank”, and “visibility”. The paper clearly and reasonably defines “rank” and “visibility”. However, it does not clearly define “relation”, and seems to flit between causal and non-causal interpretations. On page 5, the paper states “…for Q2 we want to find the **********************correlation********************** between [visibility and rank]” which is explicitly non-causal. But, on page 6, the paper says “…we want to analyse the **effect** of [rank on visibility]”. Much of the subsequent discussion then hems and haws about “confounders”, which would only be relevant for a causal version of Q2. In this discussion, the “causal effect of [time on visibility]” is explicitly considered as a confounder, which is highly suggestive of some (latent) causal version of Q2. However, the actual causal model in which a confounding variable might exist is never substantiated or stated. The paper goes on to define a somewhat kludgy procedure to control for both “timing” and “quality” as confounders. However, it is not clearly stated what the estimand is and it is thus not possible to determine whether the proposed estimator is valid or correct. Again, if Q2 was truly asking about the mere correlation between rank and visibility, the proposed estimator would be unnecessary; the paper is clearly dealing with a causal version of Q2, but not saying exactly what that causal estimand is. In summary, Q1 and Q2, as first stated, should be straightforward to answer from the survey results. However, the bulk of the paper is dedicated to a discussion of “confounders” and the derivation of an estimator that controls for preprint timing and quality. This work cannot be evaluated as correct/valid unless the causal model the paper is working with and the causal estimand that Q2 is asking about are clearly stated and defined.

The paper’s title “To ArXiv or not to ArXiv”, in combination with the survey questions, clearly (to me) imply the following causal question: does observing authors’ institutions causally affect reviewer judgements? This question seems to be what this paper really *wants* to ask. To answer this question, one would first want to confirm that reviewers actually do see arXiv preprints, which is what Q1 asks and answers (in the affirmative), and then try to assess the effect of reviewers seeing the preprint on their ultimate scores, controlling for all the confounding variables that both cause authors to upload preprints and also cause reviewers to rate them highly (e.g., paper quality). This is what the paper’s discussion on Q2 seems to hint at; however the current approach would not be valid for answering this particular question, partly because the paper defines the proxy variable for “quality” as the outcome itself.

---

### Review · Reviewer_TEsH · 2023-01-12

**Summary Of Contributions:**

The paper presents a study on quantifying the potential impact of posting scientific preprints online. The authors conducted surveys of reviewers of two top conferences (ICML and EC) and gathered their feedback on 1) if they deliberately searched for the papers they reviewed; and 2) if they have seen the papers before the review context. The authors presented results of these two questions, studied the potential associations between author/institution ranks and paper visibility, and offered a support analysis on papers that were not posted online pre-review.

I think the paper studies an interesting topic and has potentials in terms of broader impact. However, I do remain concerns about the general framing and the implications of the paper, and the current execution of the work.

**Audience:**

Yes

**Broader Impact Concerns:**

I don't see concerns as long as the surveys were conducted with proper consents

**Claims And Evidence:**

Yes

**Requested Changes:**

To summarize, I think the dataset is valuable but the paper can benefit a lot from reframing the research questions, adjusting analysis and the presentation of results accordingly.

- The introduction will need some improvements. The authors may need to consider sharpening the overall motivations, the concrete scope of the study, and the research hypotheses.

- It would be great that the analysis can be conducted to support a clear and coherent story about the relationship among variables that associated with these hypotheses. It is understandable that drawing causal relations out of observational data would be difficult, but it would be better if the authors could provide a clear picture about the variables of interests, what were controlled, and how the analysis results support the findings. Model-based statistical tests (beyond the two variable correlation test) can also be considered to address some of these concerns.

- In Figure 3, I personally find the inverted U-shape of for papers posted online and submitted to EC very interesting. Any potential explanations?

**Strengths And Weaknesses:**

Strengths

- I find the topic studied in the paper is interesting. The survey dataset the authors curated is also unique and could be impactful.

- The numbers from the "deliberately search" and the "pre-review visibility" (organically discover) questions indeed show some quantitative evidence to question the current anonymization policies.

- The authors have provided reasonable discussions about the potential confounding factors and some of them are considered in the analysis design.

Weaknesses

- My major concern is about the framing of this paper. I find it a bit difficult to justify the motivations on the two research questions proposed in the paper out of the introduction and the implications of the findings out of the analysis. For example, why pre-review visibility instead of paper acceptance is considered as outcomes in Q2. More broadly what the end outcomes are studied in this work - is it the impact of a paper (presumably more crucial for authors) or the fairness of the peer review process (perhaps more important for policymakers)? What are the research hypotheses behind the analysis? What and how the factors included in the analysis affect those outcomes? Discussions and implications in the current draft are centered around the peer review policy making, which seems not very consistent from my take from the current introduction about "help authors make informed decisions about posting preprints online".

- For the support analysis, the authors chose to use the final decision of the paper as an indicator of the paper quality, which seems built on an implicit assumption that the review process is perfect and fair. This is probably not true in most cases as the process itself will be confounded by the exposure of the paper and authors (as results in Q1). To be clear I still find the results from this analysis interesting (Fig 3) but the authors may need to be careful about the interpretations. In fact, I think it would be a good direction to push further, e.g., what would be the causal graphs among author/institution ranking, post preprints online, intrinsic quality, peer review acceptance rate (and perhaps the final paper impact/citation/etc).

- The topic studied in this work can be interesting and impactful for the computational social sciences research community, but I'm unsure if TMLR is the right audience.

---

### Decision · Action_Editors · 2023-02-26

**Recommendation:** Reject

**Comment:**

There were two issues raised by reviewers. I should note that both were initially brought up by one of the reviewers and, after considering them the two other reviewers agreed that these were major shortcomings of the current work. So the recommendation to reject the paper is unanimous. I summarize both as they were introduced by the reviewer.

The first issue is that there is no (causal) edge between quality and visibility in the causal model. This seems unnatural as high-quality papers will likely enjoy more visibility (e.g., they will be shared online more) all else being equal. Hence, it is difficult to know if the conclusions from this model hold.

The second issue is in the words of the reviewer (with light editing) "that the paper states its causal model but does not formalize what the causal estimand is (e.g., ATE, ATT). Furthermore, there is no statement (or proof) that any causal estimand is identified by the observed data, and no statement (or proof) that the proposed estimation procedure is a consistent estimate of any estimand. In short, the reader does not know the estimand, and does not know if it is being estimated."

We all agree that these are a priori difficult questions to answer and believe that if they can be addressed the study will be significant and useful to the community. I encourage the authors to consider the above in preparing the next version of their work.

I would be happy to consider a resubmission of this work to TMLR.

**Audience:**

The research question, method, dataset, and conclusions would be of great interest to the TMLR audience since arXiv is standard in the community.

**Claims And Evidence:**

The initial reviews were enthusiastic about this difficult study. They also suggested several points of possible improvement. Some of the leading suggestions were around the framing of the two research questions in the paper (Q1 and Q2) and the details of the models being used.

The authors addressed these suggestions through fairly significant revisions. In particular, the current version of the work now (clearly) poses a causal question (Q2 asks what is "the causal effect of the rank of the authors’ affiliations on the visibility of a preprint to its target audience") and provides a causal model (Fig. 1). While this stance lifts some of the ambiguity from the initial submission, reviewers unanimously agree that the causal question (Q2) is not answered to satisfaction. I provide more details below (Comments section).